# Solulan C24- and Bile Salts-Modified Niosomes for New Ciprofloxacin Mannich Base for Combatting Pseudomonas-Infected Corneal Ulcer in Rabbits

**DOI:** 10.3390/ph15010044

**Published:** 2021-12-29

**Authors:** Soad A. Mohamed, Mohamed A. Abdelgawad, Rania Alaaeldin, Zeinab Fathalla, Hossam Moharram, Raafat M. A. Abdallah, Islam M. Abdel-Rahman, Mohamed Abdel-Aziz, Gamal El-Din A. Abuo-Rahma, Mohammed M. Ghoneim, Alaa M. Hayallah, Mahmoud Elrehany, Hamdy Abdelkader

**Affiliations:** 1Department of Pharmaceutics, Faculty of Pharmacy, Deraya University, New-Minia 61519, Minya, Egypt; soad.ali@deraya.edu.eg; 2Department of Pharmaceutical Chemistry, College of Pharmacy, Jouf University, Sakaka 72341, Al Jawf, Saudi Arabia; 3Department of Biochemistry, Faculty of Pharmacy, Deraya University, New-Minia 61519, Minya, Egypt; rania_alaadin@hotmail.com (R.A.); mahmoud.elrehany@deraya.edu.eg (M.E.); 4Department of Pharmaceutics, Faculty of Pharmacy, Minia University, Minia 61519, Minya, Egypt; zeinab_minia_eg@yahoo.com; 5Department of Ophthalmology, Faculty of Medicine, Minia University, Minia 61519, Minya, Egypt; hossam.moharram@gmail.com (H.M.); raafat.abdelrahman@mu.edu.eg (R.M.A.A.); 6Department of Pharmaceutical Chemistry, Faculty of Pharmacy, Deraya University, New-Minia 61519, Minya, Egypt; islmoa2002@yahoo.com (I.M.A.-R.); gamal.aborahma@deraya.edu.eg (G.E.-D.A.A.-R.); 7Department of Medicinal Chemistry, Faculty of Pharmacy, Minia University, Minia 61519, Minya, Egypt; abulnil@hotmail.com; 8Department of Pharmacy Practice, College of Pharmacy, AlMaarefa University, Ad Diriyah 13713, Ar Riyad, Saudi Arabia; mghoneim@mcst.edu.sa; 9Department of Pharmaceutical Organic Chemistry, Faculty of Pharmacy, Assiut University, Assiut 71515, Asyut, Egypt; alaa_hayalah@yahoo.com; 10Department of Pharmaceutical Chemistry, Faculty of Pharmacy, Sphinx University, New-Assiut 71515, Asyut, Egypt; 11Department of Pharmaceutics, College of Pharmacy, King Khalid University, Abha 61441, Aseer, Saudi Arabia

**Keywords:** ciprofloxacin mannich base, keratitis, corneal ulcer, gene expression, antibiotic resistance

## Abstract

Keratitis is a global health issue that claims the eye sight of millions of people every year. Dry eye, contact lens wearing and refractive surgeries are among the most common causes. The resistance rate among fluoroquinolone antibiotics is >30%. This study aims at formulating a newly synthesized ciprofloxacin derivative (2b) niosomes and Solulan C24-, sodium cholate- and deoxycholate-modified niosomes. The prepared niosomal dispersions were characterized macroscopically and microscopically (SEM) and by percentage entrapment efficiency, in vitro release and drug release kinetics. While the inclusion of Solulan C24 produced something discoidal-shaped with a larger diameter, both cholate and deoxycholate were unsuccessful in forming niosomes dispersions. Conventional niosomes and discomes (Solulan C24-modified niosomes) were selected for further investigation. A corneal ulcer model inoculated with colonies of Pseudomonas aeruginosa in rabbits was developed to evaluate the effectiveness of keratitis treatment of the 2b-loaded niosomes and 2b-loaded discomes compared with Ciprocin^®^ (ciprofloxacin) eye drops and control 2b suspension. The histological documentation and assessment of gene expression of the inflammatory markers (IL-6, IL1B, TNFα and NF-κB) indicated that both 2b niosomes and discomes were superior treatments and can be formulated at physiological pH 7.4 compatible with the ocular surface, compared to both 2b suspension and Ciprocin^®^ eye drops.

## 1. Introduction

Bacterial keratitis is a primary cause of corneal blindness [1]. Being non-vascular, cornea receives no blood supply; therefore, the infection of the cornea can be considered a medical emergency that requires prompt action. Bacterial keratitis could progress rapidly and can cause ocular damage such as irreversible corneal melt, endophthalmitis and ultimately lead to loss of vision [1,2]. Diseases that can be associated with corneal neovascularization include inflammatory disorders, corneal graft rejection after transplantation, infectious keratitis, contact lens-related hypoxia, alkali burns, stromal ulceration or limbal stem cell deficiency [3].

The contagions responsible for bacterial keratitis are surprisingly alike worldwide; however, the predominance of each microbial etiology changes according to many factors including patients’ general health, the nature of precursor optical trauma resulting from from surgery or lesions, environmental factors and the type of the pathogen [2]. Over 1.5 million people worldwide could develop blindness from infectious corneal ulceration each year [4]. The relatively scarce attention given to infectious corneal ulceration does not reflect the impact of the condition on the most vulnerable, many of whom live in poverty.

Common causative bacteria include Staphylococcus pneumoniae and Pseudomonas aeruginosa [5]. Pseudomonas aeruginosa is the most common bacteria involved in microbial keratitis, especially among contact lens users [5,6]. Narayanan et al. showed that a history of surface eye diseases, “most notably dry eye”, was a common factor linked to bacterial keratitis. For many years, controlling of bacterial keratitis has been a diagnostic and therapeutic deadlock depending on many factors: pathogens’ susceptibility to antibiotics, topical application of verifiable antibiotics and adapting the treatment course according to the outcomes of the microbiology reports [2]. The bacterial resistance to commonly used antibiotics (fluoroquinolone and penicillin derivatives) eye drops has been reported to be very common among, as it exceeds 30% among those using in vitro antibiotic resistance profiles from ocular bacterial isolates of the different microorganisms Staphylococcus aureus, Hemophilus influenzae and Pseudomonas aeruginosa [7,8].

Infectious keratitis is a preventable and treatable ocular disease. One possible solution to infectious corneal ulceration could lie in the delivery of a simple, safe and effective community-based strategy. In addition, the early administration of topical drugs with a low level of bacterial resistance has been proven to be an effective strategy [9].

Ciprofloxacin hydrochloride (CIP) is one of fluoroquinolone antibiotics; CIP eye drops are widely prescribed by the ophthalmologists to treat bacterial keratitis and corneal ulcers [10]. CIP exerts its killing properties on susceptible bacterial strains by inhibiting the bacterial topoisomerase and gyrase enzymes required for the multiplication and growth of the bacteria [11,12]. The solubility of CIP varies according to the pH of the medium. While CIP is soluble in aqueous acid up to pH 4.5, CIP is poorly soluble at the physiological pH 7.4 and has been associated with the formation of crystalline deposits, especially from solution eye drops where the pH of the commercial Ciloxan^®^ eye drops is adjusted at pH 4.5 to ensure maximum solubility.

Upon administration to the ocular surface with pH 7.4, the disruption of the precorneal residence tear film is likely to happen, and CIP can deposit on the corneal surface, forming white deposits in corneal tissues due to poor solubility and precipitation at the physiological pH of the eye surface [13,14].

Due to the poor in vivo dissolution of ciprofloxacin topical eye drops at physiological tear fluid pH, limited penetration to eradicate Gram-positive bacteria in the anterior chamber has been recorded [15]. Other reported ocular side effects from using ciprofloxacin eye drops include burning, stinging and corneal perforation [15]. These indicate that both the ocular safety and efficacy of using ciprofloxacin eye drops need further optimization. Both chemical and formulation approaches are required to ameliorate the inherent poor solubility at the physiological pH of the tear film.

A new and promising derivative of CIP is ciprofloxacin Mannich base [16], which is superior to CIP with regard to its wide bacterial spectrum. This CIP Mannich derivative was studied on different bacterial strains and it exhibited a killing activity on Pseudomonas aeruginosa [17]. Pseudomonas aeruginosa has been proved as the main causative microorganism for keratitis. The chemical structures of CIP and ciprofloxacin Mannich derivative (2b) are shown in Figure 1.

Niosomes, the carrier of choice, are spherical vesicular aggregates of non-ionic surfactants [18]. Niosomal dispersions have favorable attributes, especially for ocular drug delivery. Span 60-based niosomes are tolerable on ocular tissues, relatively more viscous and more spreadable vehicles on the eye surface compared to aqueous buffer solutions [19].

Thus, niosomes are better off for enhancing drug permeability and prolonging ocular residence time [20]. Discomes are basically non-ionic surfactant vesicles, but they can be generated by replacing the extremely hydrophobic cholesterol molecules (the membrane stabilizer) with the water-soluble cholesteryl ether of poly (24) oxyethylene (Solulan C24). The result is non-spherical vesicles with a larger diameter known as discomes [21]. Bile salts are commonly used solubilizers and permeation enhancers [22].

Three membrane additives, two bile salts (sodium cholate and sodium deoxycholate) and Solulan C24, were investigated as bilayer membrane modifying niosomes for EE%, size, in vitro release and in vivo characteristics. The main aim of this work is to enhance the in vivo dissolution and permeation of the novel CIP derivative (2b) into the cornea via application of novel niosomal systems. In order to achieve this goal, 2b-loaded niosomes were prepared, adopting the thin film hydration method. The effect of these three membrane additives on vesicle characteristics was investigated. In addition, in vitro drug release studies across intact semipermeable membrane as well as in-vivo studies, histological documentation and gene expression of the inflammatory mediators in the cornea were conducted for the optimal niosomal evaluation.

## 2. Results and Discussion

Previous studies on ciprofloxacin indicated that the encapsulation of ciprofloxacin in liposomes has been more effective and reduced the treatment period against Bacillus anthracis [23]. Prolonging ocular release and using penetration enhancers with ciprofloxacin Carbopol and HPMC gels showed superior corneal penetration and antibacterial efficacy compared to ciprofloxacin solution eye drops [24].

### 2.1. Preparation of 2b-Loaded Niosomes an Discomes

Five different niosomal formulations were prepared, employing the thin film hydration method based on an optimized masses (mg) equivalent to the molar ratio 7:3 of the surfactant forming niosomes (Span 60) and the membrane stabilizer (cholesterol). This optimized ratio could offer residual thermal transition at the ocular surface for enhanced ocular tolerability and bioavailability [19,20].

Three different membrane additives were investigated, namely, Solulan C24, sodium cholate and sodium deoxy cholate. The inclusion of Solulan C24 into niosomes produces a more favorable version for ocular delivery compared with the conventional spherical niosomes [21]. Discomes have relatively larger size and discoidal shape that can be retained longer on the surface of the eye for better ocular bioavailability compared to conventional niosomes [21]. Sodium cholate and deoxycholate have been reported to produce a special type of surfactant vesicles called bilosomes that have desirable characteristics for enhanced permeation and skin tolerability [19].

Figure 2 shows a collective photograph of the five different prepared formulations. Apart from F3 and F4, 3 out of 5 formulations show uniform, homogeneous and milky dispersion, indicating the formation of niosomal dispersion vesicles, while F3 (containing sodium cholate) showed milky dispersions, but numerous non-hydrated large lipid aggregates were observed. On the contrary, cracked and phase separations were obtained with F4 (containing sodium deoxycholate).

These findings contradict our previous findings upon using a water-soluble drug (naltrexone hydrochloride) [25]; the results indicate that both sodium cholate and deoxycholate are poor membrane additives when used with a poorly soluble drug like 2b. The lower panel of Figure 2 shows the size distribution curves of F1 (niosomes) and F2 (discomes), with average particle sizes of 5 and 10 µm, respectively.

### 2.2. Characterization of the Prepared 2b Niosomes and Discomes

The entrapment efficiency percentages (EE%) for F1, F2 and F5 were in the following order: 91 ± 2.5%, 97 ± 4% and 81 ± 3.5%, respectively. In general, high EE% recorded for the prepared niosomal formulation could be ascribed to the hydrophobic nature of the drug; therefore, it is more happily partitioned into a surfactant bilayer than a bulk aqueous environment. Further, the EE% for discomes (F2) was superior compared to the conventional niosomes. The inclusion of cholesterol into F5 significantly affected the EE% in a negative way. For example, F5 containing 10% of cholesterol lowered the EE% by 16% compared to that of F2. Cholesterol is an extremely hydrophobic moiety that aligns itself parallel to the bilayer membrane [17]. Hence, it competed with the hydrophobic drug 2b at the limited available sites with the bilayer membranes of the vesicles.

Figure 3 demonstrates SEM micrographs for F1 (the conventional niosomes) and F2 (discomes) at 3 different magnifications: ×350, ×750 and ×1000. Spherical niosomes with smooth surfaces and an approximate diameter of 5 µm were recorded for F1. On the other hand, relatively larger sizes (>>10 µm) with discoidal shaped vesicles and irregular surfaces were recorded for F2. These findings confirm the formation of niosomes and discomes for F1 and F2, respectively.

From SEM and EE% studies, discomes with larger sizes and irregular surfaces could stay longer on the surface of the eye compared to small and spherical vesicles. These lend discomes more favorable features for ocular delivery than the conventional niosomes [21,26].

Figure 4 shows different release profiles for 2b from the three selected niosomes and were compared with that for control 2b suspension formulation. The lowest release rate (1 %.h^−1^ and extent (less than 5%) of cumulative drug release were recorded with the control 2b suspension formulation. On the other hand, all of the selected niosomal formulations (F1, F2 and F5) showed significantly higher drug release rates and extents (from approx. 40% to 80%). These results could be assigned to the availability of 2b in soluble forms in the bilayer membranes of niosomes and are ready to partition; however, more time was required for the solubility and dissolution of drug particles in the control suspension formulation. Among the selected three different niosomal formulations, the typical spherical niosomal formulation F1 showed the highest drug release compared to discomes F2 and F5. The rigidity of the bilayer membrane of conventional niosomes that included the highest percentage of cholesterol (the extremely hydrophobic and rigid molecule) could be the possible reason of ejecting, i.e., ‘’kicking’’, the hydrophobic drug molecules out of the bilayer membranes [27].

Similar results were reported elsewhere. Carvedilol niosomes composed of Span 60:Tween 60:cholesterol at a 25:25:50 ratio showed faster release rates compared to those composed of a lower ratio of cholesterol (Span 60:Tween 60:cholesterol of 35:35:30) [28].

Six release kinetics models were studied to understand the drug release mechanisms from the selected formulations. The regression coefficients and release rate constants estimated from the used models were demonstrated in Table 1. The results showed that the best fitting model, as indicated from r values > 0.99, was the Higuchi model/diffusion release mechanism from the vesicles. The release rate constants estimated for F1, F2 and F3 by the Higuchi model were 4.3, 2.5 and 2.3 %.min^−0.5^, respectively.

Release rate constants were in good accordance with the release profiles (Figure 4). F5 recorded q slightly lower release rate compared to that for F2. However, there were no statistically significant (*p* > 0.5) differences between the release rate constants for F2 and F5. The main difference in the composition of the two discomes formulations was that F5 contained 10% of cholesterol. This cholesterol level slightly decreased the release rate but was not significant. Therefore, both F1 (niosomes) and F2 (discomes) were selected for further in vivo studies.

### 2.3. In Vivo Study

The included rabbits’ corneas in each group were evaluated over 4 days from inducing corneal ulcer, inoculating with Pseudomonas aeruginosa and starting the treatment 24 h post-infection (Figure 5). The corneas were photographed and the size of the ulcer was evaluated by measuring the ulcer area in mm^2^. The measurements were evaluated and analyzed with ImageJ software, with 64-bit Java 1.8.0_172.

In group A (the untreated group), the size of the ulcer progressively increased and the cornea became totally opacified within 4 days of the follow up. The ulcer size ranged between 8.23 and 13 mm^2^ (the mean ulcer size was 10.878 ± 2.10 mm^2^).

In group B (treated with Ciprocin^®^ eye drops), the corneal ulcer gradually improved and the corneas became clear by the fourth day of follow up. The size of the ulcer ranged between 8.93 mm^2^ before treatment and 0 after treatment (mean 4.8276 ± 3.43 mm^2^). In group C (treated with 2b suspension), the ulcer progressively increased and the corneas became totally affected by the end of follow up.

The ulcer size ranged between 8.87 and 13 mm^2^ (mean 10.8924 ± 1.35 mm^2^). In group D (treated with niosomes), the ulcer gradually improved and decreased in size, with total improvement by the fourth day. The size of the ulcer ranged between 0 (after treatment) and 8.77 mm^2^ before treatment (mean 5.1792 ± 3.09 mm^2^).

In group E (treated with discomes), the ulcer decreased in size and complete improvement was evident in the 3rd day of follow up. The size of the ulcer ranged between 0 (before treatment) and 8.78 mm^2^ after treatment (mean 5.6576 ± 3.27 mm^2^). Group E (treated with discomes) showed marked improvement on the 2nd day of treatment in comparison to other groups that showed improvement. Moreover, the complete resolution of the ulcer area was reached in group E on the 3rd day in comparison to the 4th day in groups B & D. Discomes have more ocularly favorable characteristics than conventional niosomes. This is due to them having larger sizes and non-spherical (discoidal) shapes that can resist tears flushing and stay longer on the surface of the eye compared with the smaller size and spherical-shaped niosomes [21,26].

### 2.4. Histological Documentation of the Rabbits’ Corneas

Figure 6a shows histological micrographs of normal cornea; the outer covering is non-keratinized stratified squamous epithelium (star), the underlying Bowman’s membrane (asterisk), the stromal collagen fibers (red arrows) and keratocytes (black arrows). Figure 6b,c shows infected cornea, intrastromal bacterial colonies (circle), dilated and congested blood capillaries (black arrows), inflammatory polymorphonuclear cells infiltration (yellow arrows) and disorganized collagen bundles (asterisks). Hydropic swelling of some of the covering epithelial cell and basement membrane was absent and replaced by inflammatory cells. Figure 6c shows corneal ulcer (arrow head) and stromal neutrophil cells infiltration (inset).

Figure 7A shows the synthetic drug suspension-treated group; the corneal epithelium appeared as multi-layered squamous-like cells, the basal and supra-basal layers show hydropic swelling and others showed apoptosis (blue and red arrows). The stroma shows fewer inflammatory cells infiltrate (circle), moderate edema (rectangle) and minor disorganized collagen (asterisk). Figure 7C shows the niosomes-treated group; apoptotic and mitotic figures (red & black arrows) are seen among the hyperplastic epithelium (star). The stromal edema is very minimal (blue arrows), and the Descemet’s membrane (DM) is thick.

Figure 7B shows the commercial ciprofloxacin (Ciprocin^®^) eye drops-treated group; the epithelium is about five layers thick (star), the Bowman’s membrane is attenuated at certain sites (thick arrow), the stroma shows fibroblast cell proliferation (inset) and the DM has normal thickness. Figure 7D shows the discomes-treated group, normal epithelium (star) and intact Bowman’s membrane (blue arrow). The corneal stroma shows no edema, normal keratocytes (black arrows), well-organized collagen fibers and DM with normal thickness.

In addition to the superior healing rates recorded for both 2b niosomes and 2b discomes, histological documentation could indicate that niosomes and discomes drug delivery systems are well-tolerated, effective and safe carriers for 2b. This is because the normal structure of the cornea was obtained even after complete wound healing. These results are in favour of the good ocular tolerability and safety of the prepared niosomes and discomes.

### 2.5. Gene Expression of IL-6, IL-1B, TNFα, and NF-κB

Keratitis is the inflammation of the cornea. NF-κB plays a central role in the induction and regulation of pro-inflammatory cytokines in both innate and adaptive immune response [29,30]. In response to inflammation, macrophages become activated and secrete the cytokines (e.g., IL-1B, IL-6 and TNFα) [31].

IL-6 and IL-1B are pleiotropic cytokines that mediate the inflammation in tissues and organs including the cornea. TNF-α showed proinflammatory and immunoregulatory characteristics, for instance, and TNFα exhibited immunoregulatory activity on the differentiation of B-cells, T cells and dendritic cells. The expression of IL-6, IL1B, TNFα and NF-κB was investigated in the present study.

The mRNA levels of IL-6, IL1B, TNFα and NF-κB in the negative control, untreated positive control and treated groups are shown in Figure 8. The quantitative real-time findings revealed that all of the treated groups demonstrated statistically significant (*p* < 0.05) lower expressions of the four inflammatory mediators measured. Ciprocin^®^ eye drops significantly (*p* < 0.05) lowered the mRNA levels of NF-κB and IL-6 when compared to the infected untreated animals.

Also, the marketed ciprofloxacin eye drops showed a notable (*p* < 0.01) decrease in TNFα when compared to infected untreated animals. While Ciprocin^®^ eye drops showed non-significant (*p* > 0.05) activity on IL-1B. The drug exerted a significant (*p* < 0.05) decrease on NF-κB, a notable (*p* < 0.01) decrease on TNFα and non-significant (*p* > 0.05) activity on IL-6 and IL-1B when compared to infected untreated animals. 2b niosomes exerted significant (*p* < 0.05) inhibition on IL-1B and notable (*p* < 0.01) decreases of the mRNA levels of IL-6, TNFα and NF-κB when compared to infected untreated animals. 2b discomes exhibited significant (*p* < 0.01) inhibition on IL-6, IL-1B and NF-κB and substantial inhibition on TNFα gene expression when compared to infected untreated animals, as shown in Figure 8.

Since 2b discomes formulation showed better characterization in the above-mentioned findings, the significance of 2b discomes (F2) on the modulation of gene expression against 2b niosomes (F1), 2b suspensions and Ciprocin^®^ eye drops was compared. There was no significant (*p* > 0.05) difference in different gene expression between discomes and niosomes formulations. However, discomes (F1) exerted lower (*p* < 0.01) mRNA levels in TNFα and NF-κB, and notable decreases (*p* < 0.001) in IL-6 gene expression, when compared to the 2b suspension, while discomes exerted a significant (*p* < 0.01) decrease in IL-6 gene expression when compared to Ciprocin^®^ eye drops.

## 3. Materials and Methods

Sodium cholate, sodium deoxycholate, cholesterol and cellulose membrane (molecular weight cut-off 12,000–14,000 Da) were obtained from Sigma–Aldrich Chemical Co. (Poole, UK). Solulan C24 was donated from Lubrizol, France. Sodium cholate and Span 60 were purchased from Fisher Chemical, (Loughborough, Leicestershire, UK).

### 3.1. Chemistry

We have recently published a new ciprofloxacin-Mannich derivative that has been given the abbreviation (2b). 2b was synthesized at the organic chemistry lab, Faculty of pharmacy, Deraya University according to the previously published method [16]. The selected compound (2b) was synthesized in good yield through Mannich reaction between ciprofloxacin and 2-naphthol by refluxing the two components with formaldehyde in ethanol. The compound was confirmed and checked by 1H- NMR &13C-NMR spectral data as well as Mass spectrometry and elemental analysis.

### 3.2. Bacteria

Pseudomonas aeruginosa bacteria were grown from the Persian-type culture. The bacteria were sub-cultured using Muller- Hinton agar plates and incubated at 37 °C. Pseudomonas aeruginosa PAO1 was obtained from the Microbiology Department, Deraya University.

### 3.3. Methods

#### 3.3.1. Preparation of 2b-Loaded Niosomes and Discomes

Various 2b-loaded niosomes based on the design shown in Table 2 were generated using the thin film hydration method. Span 60 was the surfactant-forming vesicles; bilayer membrane additives, namely, cholesterol, Solulan C24, sodium cholate and sodium deoxycholate were used [25]. The drug, surfactant and membrane additives were transferred into a 50 mL flask with a round bottom. The surfactant/lipid powder was dissolved in 5 mL of methanol:chloroform mixture (1:1 *v*/*v*). The solvent was rotary evaporated under a vacuum, and the formed lipid film on the bottom of the flask was dispersed in 10 mL of phosphate buffer saline of pH 7.4 for 60 min at 60 °C.

#### 3.3.2. Characterization of Synthesized CIP-Loaded Niosomes

##### Entrapment Efficiency (EE) %

The EE% of niosomes for 2b was determined by centrifugation. The niosomes were spun at 5000 rpm for 1 h (h). Thereafter, the drug amount in the supernatant was determined using an ultraviolet-visible spectrophotometer (JENWAY, Shanghai, China) at a wavelength of 272 nm. The EE% of 2b niosomes was estimated through Equation (1):(1)$$\mathrm{EE}\%=\frac{W\;\mathrm{initial}\;\mathrm{drug}-\;W\;\mathrm{drug}\;\mathrm{in}\;\mathrm{supernatant}}{W\;\mathrm{initial}\;\mathrm{drug}}\times 100$$
where W denotes the mass (mg) of drug.

##### Scanning Electron Microscope (SEM)

The size and morphology of some selected niosomes were studied using the SEM (KYKY EM 3200, China). The prepared niosomes were diluted (1:10) with deionized water. The diluted niosomes were dropped on an amorphous polycarbonate grid, allowed to dry and were sputter coated with gold and examined with accelerating voltage at 200 kV.

##### In Vitro Release

In vitro release was studied using in-house modified-Franz diffusion cells. The receptor compartment was thermostated by immersion in a shaking water bath (MESB-1A, Labomiz, Scientific limited, Farmingdale, NY, USA) at 50 strokes per minute and adjusted at 35 ± 0.5 °C. The receptor compartment (50 mL) was filled with the phosphate buffer saline containing 0.1% Tween 80 (pH 7.4) and constantly stirred by small magnetic bars.

A dialysis membrane with MW of 12,000–14,000 Dalton separated the donor compartment and receptor tube. One mL of each niosomal dispersion or 2b suspension was transferred into the donor compartment; one mL was withdrawn from receptor compartment at specified time points: 15, 30, 60, 120, 180, 240 and 300 min. The experiment was repeated thrice, and the cumulative drug release was analyzed spectrophotometrically at 272 nm, as described above.

##### Release Kinetics

Release data were fitted into different mathematical models (Equations (2)–(7)) for studying the release mechanisms from the selected niosomes [32]:

Zero order:(2)$$Q_{t}=\;Q_{o}+K_{o}t$$

First order:(3)$$\frac{dQt}{dt}=K_{1}Q_{o}$$

Equation (3) can be simplified as:(4)$$logQ_{t}=logQ_{o}-\frac{tK1}{2.303}$$
where *Q_t_* and *Q_o_* are the amounts of drug released at time *t* and time = 0, respectively; *K_o_* and *K*_1_ are the zero order and first order rate constants, respectively.

Higuchi diffusion:(5)$$Q=K_{H}t^{0.5}$$
where *k_H_* is the release constant

Hixon Crowell:(6)$$Q_{o}^{1/3}-Q_{t}^{\frac{1}{3}}=\;tK_{HC}\;$$
where *k_HC_* is the Hixon-Crowell release rate constant.

Baker–Lonsdale:(7)$$\frac{2}{3}\left[1-{\left(1-\frac{Mt}{M\infty }\right)}^{2/3}\right]-\frac{Mt}{M}=\;tK_{3}$$

Korsmeyer–Peppas:(8)$$\frac{Mt}{M\infty }=k\;t^{n}$$
whereas *Mt/M∞* refers to the percentage cumulative drug release at t; *n* is the release exponent.

##### In Vivo Studies

Preparation of bacteria

Pseudomonas aeruginosa PAO1 was obtained from Deraya University. Muller Hinton agar (Merck, Germany) was employed for growing the bacteria. The concentration of bacterial suspension was adjusted to 105 CFU mL^−1^.

Animals and grouping

Twenty-five New Zealand rabbits weighting 1.5–2.5 kg were obtained from the animal house (Deraya University). The rabbits were divided into four groups (5 rabbits for each group). Each left eye was treated as follows.

Group A was negative control (no infection; no treatment); group B was positive control and treated with Ciprocin^®^ 0.3% eye drops; group C was positive control and treated with 2b 0.3 suspension; group D was positive control and treated with F1 niosomes; and group E was positive control and treated with F2 discomes. The right eyes were left untreated as positive untreated control.

The corneas were photographed, and the size of the ulcer was evaluated by measuring the ulcer area in mm^2^. The measurements were evaluated and analyzed with the ImageJ software. This in vivo study was approved by the Ethical Review Board of Faculty of Pharmacy (Approval no. 2/2021), Deraya University. The study protocol complied with all national and international guidelines for the care and the use of animals.

Induction of experimental pseudomonas-infected corneal ulcer (compounded bacterial keratitis)

Corneal ulcers in rabbits were induced in both eyes, as previously mentioned, using alcohol 70% *v/v* [33]. The surgically induced corneal epithelial ulcers were infected with the spreading one CFU of Pseudomonas aeruginosa on the affected ulcer lesion. The treatment was initiated 24 h post-ulcer formation and bacterial inoculation to allow for bacterial virulence and keratitis induction. Each left eye was treated, as mentioned above, three times daily by a single instillation of drop of each treatment for four days. The rabbits’ eyes were stained with sodium fluorescein solution (2% *w*/*v*) for ulcer assessment purposes and then photographed. At the end of the 4-day experimental period, the animals were sacrificed and the corneas were dissected for subsequent histological examination in 10% formalin and molecular analysis stored at −80 °C.

##### Histology and Immunohistochemistry

The affected corneal tissues of rabbits’ eyes from the in vivo studies were excised and placed in 10% formalin in PBS pH 7. 5 and dehydrated using absolute ethanol. Small sections (5 mm) of the fixed corneal tissues were sectioned and H&E stained for 5 min. The histological sections were visualized using the Olympus Biological Trinocular microscope, Model: CX31, Tokyo, Japan and connected With HD digital Camera model: XCAM, Tokyo, Japan (HD5mega pixel).

##### RNA Isolation and qPCR Assay

RNA extraction was performed for all tissue homogenate using Branson digital sonicator (Emerson, St. Louis, MO, USA) with TRIzol (ThermoFisher Scientific, Inc., Waltham, MA). The samples were treated with DNase I; cDNA synthesis was achieved with cDNA Reverse Transcription Kit (ThermoFisher Scientific, Inc., Waltham, MA, USA). The expression of IL-6, IL-1B, TNFα and NF-κB genes was assessed by real-time qPCR. Glyceraldehyde 3-phosphate dehydrogenase (GAPDH) was used as a control [34]. The sequences of the primers, National Center for Biotechnology Information, are mentioned in Table 3. The quantification of mRNA was achieved using Biosystems StepOne TM PCR (ThermoFisher, Waltham, MA, USA).

Triplicate RT-PCR reactions were performed for each sample. Cycle threshold (CT) was determined in treated cells relative to untreated ones, and their respective *CT* values were estimated through Equations (9)–(11) [35].
(9)$$\Delta CT=CT_{target\;gene}-CT_{reference\;gene}$$
(10)$$\Delta \Delta CT=\Delta CT_{treated\;sample}-\Delta CT_{untreated\;control}$$
(11)$$Ratio=2^{\Delta \Delta CT}$$

The CT value obtained the gene of interest from the internal control GAPDH (Δ*CT* = *CT*, *target—CT*, control). To exclude the generation of non-specific compounds and to characterize the obtained amplified mixture with the avoidance of contamination, a melting curve analysis was achieved between 60–95 °C at 1 °C intervals with the Rotor-Gene 6000 Series Software 1.7 (QIAGEN, the Netherlands) using the SYBR Green fluorescent dye.

### 3.4. Statistical Analysis

Significant differences among different formulations and animal groups were analyzed by one-way ANOVA using GraphPad Prism 5.03 software, San Diego, CA, USA.

## 4. Conclusions

Antibiotic resistance among the isolates of bacterial keratitis alarmingly increases and is now routinely reported by the ophthalmologists. In addition, the epidemiology of bacterial keratitis is spirally on the rise due to increasing demand on refractive surgeries, increasing of number of dry eye patients and contact lens users. A newly synthetized ciprofloxacin derivative based on Mannich base (2b) has been reported to show antimicrobial activities. Three membrane additives, two bile salts (sodium cholate and deoxycholate) and a non-ionic surfactant, were investigated to modify bilayer membranes’ vesicles. Among the three additives used, only Solulan C24 produced a special version of non-spherical niosomes called discomes.

A Pseudomonas aeruginosa-infectious ulcerative corneal model in rabbits was employed to study the antibacterial activity of 2b-loaded niosomes and discomes. The antibacterial activities and corneal wound healing rates of niosomes and discomes were compared with 2b suspension and Ciporcin eye drops. The histological documentation and gene expression of inflammatory markers IL-6, IL1B, TNFα and NF-κB indicated the superior antibacterial and complete healing of the prepared 2b niosomes and discomes.

The developed niosomes and discomes were tolerable and effective, as they were formulated at the physiological pH 7.4, compared to Ciprocin^®^ (ciprofloxacin) eye drops, which were formulated at low pH 4.5 to ensure complete solubility. These findings warrant the use of the newly synthetized ciprofloxacin derivative for the treatment of Pseudomonas aeruginosa keratitis.

## Figures and Tables

**Figure 1 pharmaceuticals-15-00044-f001:**
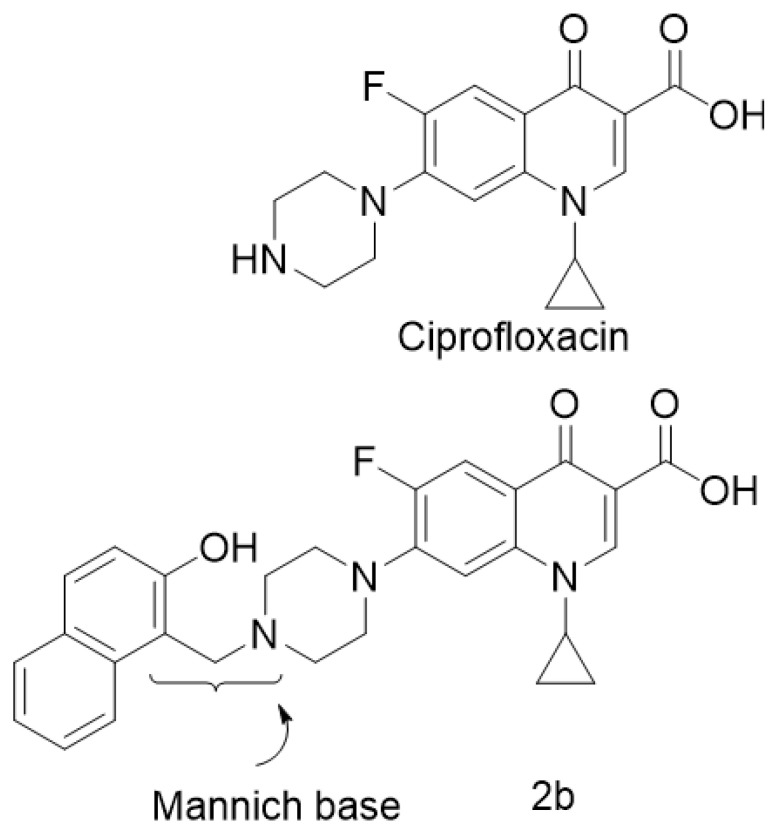
The chemical structure of ciprofloxacin (CIP) and its Mannich base derivative (2b).

**Figure 2 pharmaceuticals-15-00044-f002:**
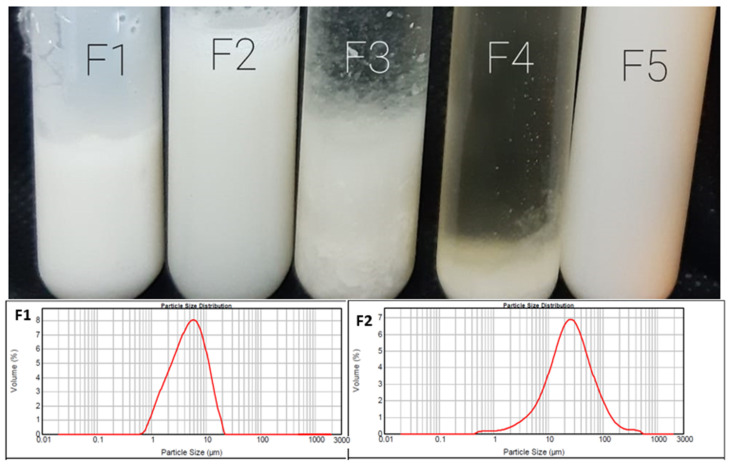
Photographs of the five different niosomal preparations (**the top panel**) and size distribution of some selected niosmes (F1) and discomes (F2) (**the bottom panel**).

**Figure 3 pharmaceuticals-15-00044-f003:**
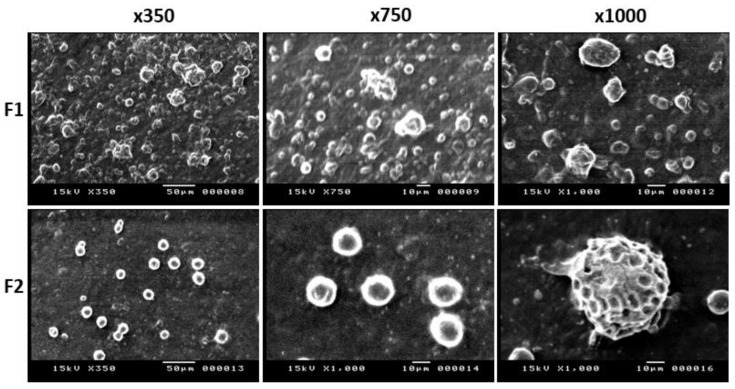
Scanning electron micrographs for F1 (conventional niosomes) and F2 (discomes) at different magnifications.

**Figure 4 pharmaceuticals-15-00044-f004:**
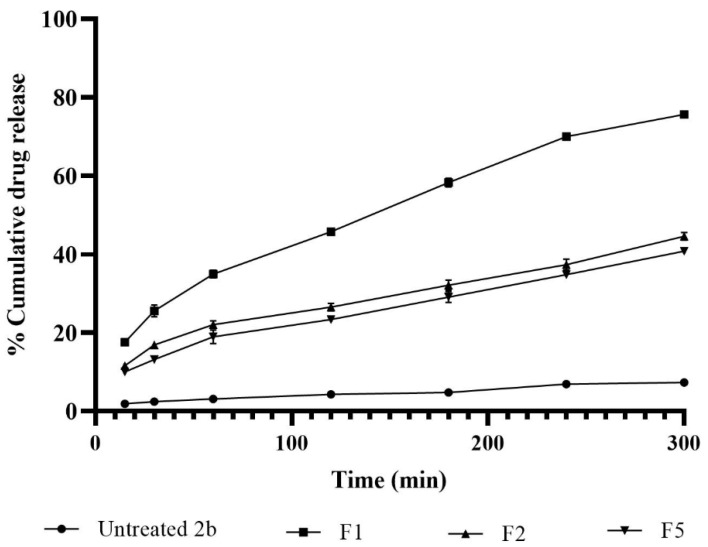
In vitro release of the ciprofloxacin Mannich derivative (2b) from some selected niosomal formulations and control 2b suspension.

**Figure 5 pharmaceuticals-15-00044-f005:**
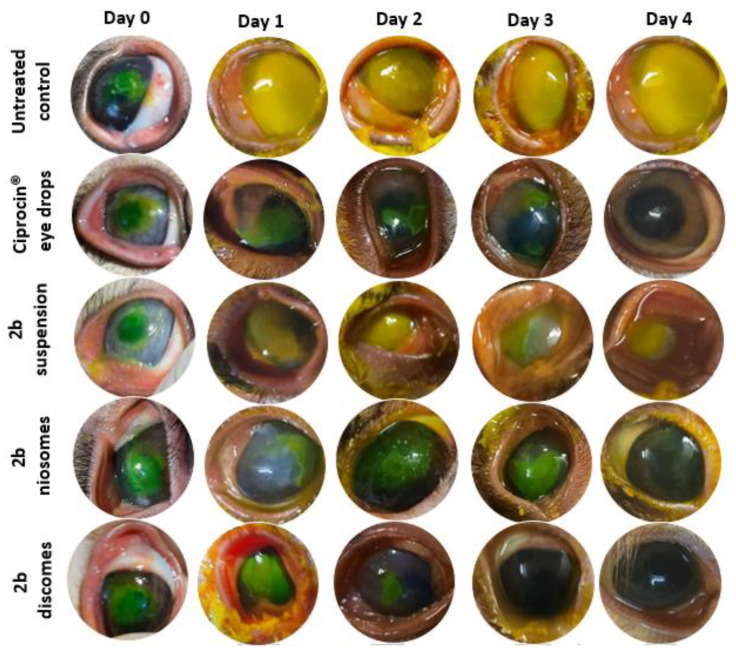
Photographic documentation of fluorescein-stained Pseudomonas infected corneal ulcers in rabbit eyes for untreated and treated groups with commercial eye drops, suspension.

**Figure 6 pharmaceuticals-15-00044-f006:**
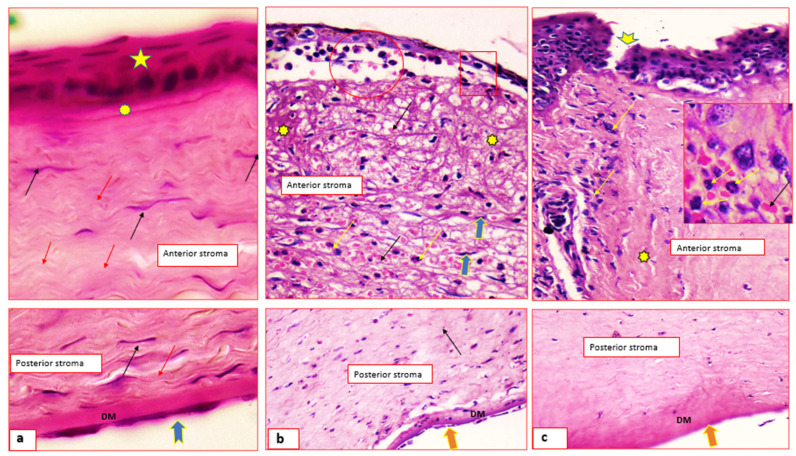
Histological micrographs of the excised rabbits’ corneas for negative untreated control (**a**) and untreated positive (infected) control (**b**,**c**), hematoxylin and eosin stain × 200 & 400.

**Figure 7 pharmaceuticals-15-00044-f007:**
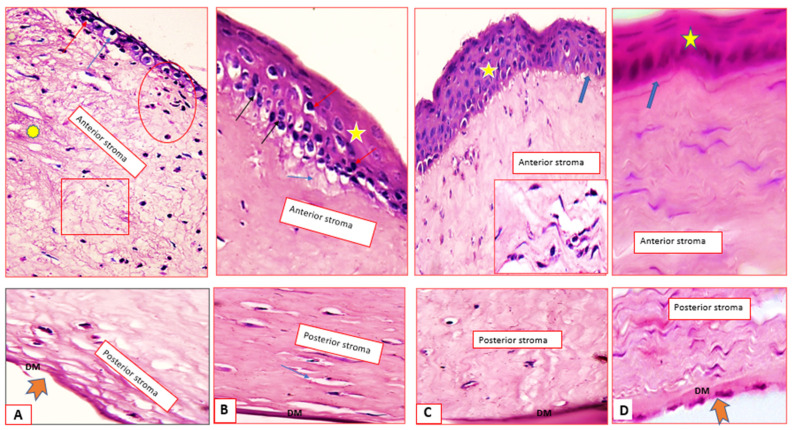
Histological micrographs of the excised rabbits’ corneas for 2b suspension (**A**), Ciprocin^®^ eye drops (**B**), niosomes (**C**), discomes (**D**) and hematoxylin and eosin stain × 200 & 400.

**Figure 8 pharmaceuticals-15-00044-f008:**
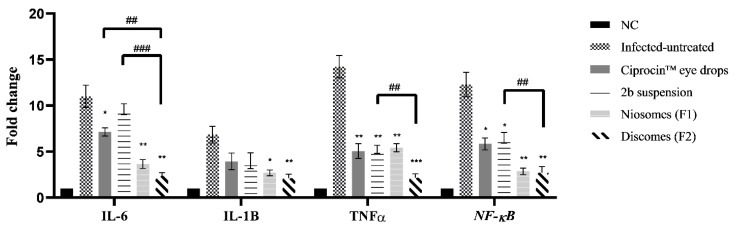
Expression of IL-6, IL1B, TNFα and NF-κB in normal untreated control (NC) and different groups after treatments. Data presented as means ± SEM. Significant differences were analyzed through one-way ANOVA. Where * *p* < 0.05; ** *p* < 0.01; *** *p* < 0.001, compared to infected untreated group and ## *p* < 0.01; ### *p* < 0.001, compared to discomes (F2).

**Table 1 pharmaceuticals-15-00044-t001:** Regression coefficient (R) and release rate constant (K) generated from different kinetics models.

Formula	Zero		First		Higuchi		Hixon-Crowel		Baker & Lonsdal		Korysmayer-Peppas	
	R	K_0_	R	K	R	K_H_	R	K_HC_	R	K_3_	R	*n*
F1	0.79	0.72	−0.79	−1.6	0.999	4.3	0.79	0.72	0.98	0.0004	0.99	0.5
F2	0.8	0.74	−0.81	−1.7	0.995	2.5	0.81	0.74	0.987	0.0001	0.98	0.4
F5	0.81	0.74	−0.81	−1.7	0.999	2.3	0.81	0.74	0.993	0.0001	0.98	0.44

**Table 2 pharmaceuticals-15-00044-t002:** Composition (molar ratio) of various 2b-loaded niosomes.

Formulation	Drug (%)	Span 60	Cholesterol	Solulan C24	Sodium Cholate	Sodium Deoxycholate
F1	0.3	7	3	-	-	-
F2	0.3	9	-	1	-	-
F3	0.3	9	-	-	1	-
F4	0.3	8	1	-	-	1
F5	0.3	8	1	1	-	-

**Table 3 pharmaceuticals-15-00044-t003:** Sequences of the primers generated from NCBI.

Primer	Sequence of the Primer
IL6	Forward: 5’-GGCACTGGCGGAAGTCAATC-3′ Reverse: 5′-ACTCCATCAGCCCCGAAGTG-3′
IL-1B	Forward: 5′-AGC TTC TCC AGA GCC ACA AC-3 Reverse: 5′-CCT GAC TAC CTT CAC GCA CC-3′
TNFα	Forward: 5′-GAG AAC CCC ACG GCT AGA TG-3′ Reverse: 5′-TTC TCC AAC TGG AAG ACG CC-3′
NF-κB	Forward: 5′-TGGGGACAGCGTCTTACACC-3′ Reverse: 5′-TGCCAAGTGCAAGGGTGTCT-3′
GAPDH	Forward: 5′-GTC AAG GCT GAG AAC GGG AA-3′ Reverse: 5′-ACA AGA GAG TTG GCT GGG TG-3′

## Data Availability

Data is contained within the article.

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
