# Peer review of "Solulan C24- and Bile Salts-Modified Niosomes for New Ciprofloxacin Mannich Base for Combatting Pseudomonas-Infected Corneal Ulcer in Rabbits"

_pharmaceuticals, 2021, doi:10.3390/ph15010044_

Round 1

Reviewer 1 Report

The article : "Solulan C24- and bile salts-modified niosomes for new ciprofloxacin Mannich base for combatting Pseudomonas-infected corneal ulcer in rabbits" provides some interesting aspects regarding this potential formulation. The methods and results seem appropriate and sufficient to provide some detailed aspects about the formulation.

However, some changes should be made to enhance the mansucript : 

  • firstly, english should be checked throughout the manuscript. It is important that the manuscript is throroughly checked prior submission. For instance, I did not understand the first sentence of the abstract and the lines 105 to 107 should be removed from the manuscript
  • abstract and introduction : 

- reading both the abstract as well as the introduction of the manuscript, I did not clearly perceive the context of the formulation. The authors should clearly state which are the critical quality attributes (e.g. drug release, drug availability, tolerance...) and the targeted product profile.

- regarding the resistance to Pseudomonas, the author may discuss of such cases in the context of the use of eye drops : this may help to directly understand why the currently marketed eye drops do not respond to the need

- regarding toxicity, it should also be highlighted if there are reported cases of toxicity of ciprofloxacine eye drops

  • Results and discussion

-Prior to give the results and discussion of the formulations,  the authors should provide some information on the currently existing formulation, including that reported in research papers

- Though the results show that the formulation may be more safe and efficient than what is marketed, the results and discussion part clearly lacks a discussion of the results. For instance, lines 161-163 :  "Among the selected three different niosomal formulations, the typical spherical nio
somal formulation F1 showed the highest drug release, compared to discomes F2 and F3.". This is important to find some elements which may explain these result, and not by use of only self citation ([19]). For other parts of the manuscript you should maybe discuss on the previous results that were obtained for other ciprofoloxacine eye drops ; please avoid only self citation for the discussion

  • Finally, it would be appropriate to have a concluding paragraph for each of the R & D chapters as some of them are very descriptive

Author Response

Reviewer 1

The article : "Solulan C24- and bile salts-modified niosomes for new ciprofloxacin Mannich base for combatting Pseudomonas-infected corneal ulcer in rabbits" provides some interesting aspects regarding this potential formulation. The methods and results seem appropriate and sufficient to provide some detailed aspects about the formulation.

However, some changes should be made to enhance the mansucript : 

  • firstly, english should be checked throughout the manuscript. It is important that the manuscript is throroughly checked prior submission. For instance, I did not understand the first sentence of the abstract and the lines 105 to 107 should be removed from the manuscript

Lines 105 and 107 were removed and the whole manuscript has been rechecked for grammar and spelling g mistakes.

  • abstract and introduction : 

- reading both the abstract as well as the introduction of the manuscript, I did not clearly perceive the context of the formulation. The authors should clearly state which are the critical quality attributes (e.g. drug release, drug availability, tolerance...) and the targeted product profile.

Both abstract and the introduction have been modified to highlight the aim and specific objectives of the study.

- regarding the resistance to Pseudomonas, the author may discuss of such cases in the context of the use of eye drops : this may help to directly understand why the currently marketed eye drops do not respond to the need.

The introduction has now been modified accordingly to justify the need for the new ciprofloxacin derivative and bactwrial resistance reported from fluorquinolone eye drops.

- regarding toxicity, it should also be highlighted if there are reported cases of toxicity of ciprofloxacine eye drops

The toxicity of ciprofloxacin eye drops has now been provided.

  • Results and discussion

-Prior to give the results and discussion of the formulations,  the authors should provide some information on the currently existing formulation, including that reported in research papers.

Some examples on ciprofloxacin formulations from the literature have now been provided.

- Though the results show that the formulation may be more safe and efficient than what is marketed, the results and discussion part clearly lacks a discussion of the results. For instance, lines 161-163 :  "Among the selected three different niosomal formulations, the typical spherical nio
somal formulation F1 showed the highest drug release, compared to discomes F2 and F3.". This is important to find some elements which may explain these result, and not by use of only self citation ([19]). For other parts of the manuscript you should maybe discuss on the previous results that were obtained for other ciprofoloxacine eye drops ; please avoid only self citation for the discussion

The results and discussion section has now been modified accordingly.

  • Finally, it would be appropriate to have a concluding paragraph for each of the R & D chapters as some of them are very descriptive

Conclusive remarks were added at the end of each R and D section

Reviewer 2 Report

Authors proposed a paper entitled “Solulan C24- and bile salts-modified niosomes for new ciprofloxacin Mannich base for combatting Pseudomonas-infected corneal ulcer in rabbits” for the publication in Pharmaceutics, MDPI.

The paper has a good scientific soundness, but some information is missing. For example, I would suggest adding a particle size distribution diagram of the produced niosomes.

Moreover, I would add to the Introduction section, a sentence of comparison among the use of liposomes and the use of niosomes, with proper references.

Abstract

“entrapment efficiency %,” I would say “percentage”

Line 40. “IL-6, IL1B, TNF and NF”. double space.

Introduction

Line 48. “Bacterial keratitis is a primary cause of corneal blindness”. add reference.

Line 58. “most notably dry eye” I would suggest to include this into parenthesis. And remove the double spaces.

Line 63. “The bacterial resistance to”. how was the bacterial resistance quantified in terms of percentage?

Line 66. “CIP”. please add an abbreviation list, according to the journal guidelines.

Line 71. “CIP is poorly soluble at the physiological pH 7.4” do you have significant literature references to support this sentence?

Line 78. “promising derivative of CIP derivative is a ciprofloxacin” derivative of derivative?

Line 84. “cholesterol” the addition of cholesterol is not compulsory in the formation of niosomes.

Line 86. “compared to aqueous buffer solutions” add a reference here, please.

Line 97. “thin film hydration method”. therefore, niosomes are created with Bangham method?

Results

Line 105. “This section may be divided by subheadings. It should provide a concise and precise description of the experimental results, their interpretation, as well as the experimental conclusions that can be drawn.” this becomes to the template of this kind of paper and should be removed.

Line 110. “molar ratio 7:3” on mass basis? specify

Line 134. “91% ± 2.5, 97% ± 4 and 81%± 3.5 respectively” authors could organize their results in terms of encapsulation efficiency, mean size and zeta potential, using a table.

Figure 3 describes the shape of niosomes and gives an idea of their size and mean dimensions. However, could author indicate mean size and standard deviation for each sample produced?

Line 245. “TNF and NF-κB” a space more is present here.

Line 251. “TNF, when comp” same problem

Line 253. “on TNF and non-significant” same problem

Line 258. “on TNF gene expression” same.

Materials and methods

Line 279. “recently published a new ciprofloxacin-Mannich derivative” why published? maybe “formulated”, “produced”?

Line 310 “W denotes the amount of drug.” should be “the mass of drug”. It would be better to use concentration in this formula

Line 327 “In vitro release”. In vitro should be in italique.

Line 335. this equation should become Nr. 4 and be put into the center.

Equation 4 should become equation 5 and so on.

Is it compulsory to indicate Section Nr. 5 on patents?

Author Response

Reviewer 2

Authors proposed a paper entitled “Solulan C24- and bile salts-modified niosomes for new ciprofloxacin Mannich base for combatting Pseudomonas-infected corneal ulcer in rabbits” for the publication in Pharmaceutics, MDPI.

The paper has a good scientific soundness, but some information is missing. For example, I would suggest adding a particle size distribution diagram of the produced niosomes.

Size distribution graphs have been provided (Figure 2)

Moreover, I would add to the Introduction section, a sentence of comparison among the use of liposomes and the use of niosomes, with proper references.

 Comparison among liposomes and niosomes have now been provided at the start of results and discussion section. In addition the introduction has now been modified accordingly.

Abstract

“entrapment efficiency %,” I would say “percentage”

Percentage has now been modified accordingly

Line 40. “IL-6, IL1B, TNFa and NF”. double space.

The line space has now been adjusted.

Introduction

Line 48. “Bacterial keratitis is a primary cause of corneal blindness”. add reference.

A reference has been provided.

Line 58. “most notably dry eye” I would suggest to include this into parenthesis. And remove the double spaces.

Parenthesis was added.

Line 63. “The bacterial resistance to”. how was the bacterial resistance quantified in terms of percentage?

The sentence has now been clarified further and modified accordingly

Line 66. “CIP”. please add an abbreviation list, according to the journal guidelines.

List of abbreviation has now been provided

Line 71. “CIP is poorly soluble at the physiological pH 7.4” do you have significant literature references to support this sentence?

More references were added to support this idea in the introduction section (lines 80 to 88).

Line 78. “promising derivative of CIP derivative is a ciprofloxacin” derivative of derivative?

This is a valid point. The sentence has now been corrected.

Line 84. “cholesterol” the addition of cholesterol is not compulsory in the formation of niosomes.

The word cholesterol has now been removed.

Line 86. “compared to aqueous buffer solutions” add a reference here, please.

Reference 17 has now been provided

Line 97. “thin film hydration method”. therefore, niosomes are created with Bangham method?

Yes, it is the method mentioned with liposomes formation.

Results

Line 105. “This section may be divided by subheadings. It should provide a concise and precise description of the experimental results, their interpretation, as well as the experimental conclusions that can be drawn.” this becomes to the template of this kind of paper and should be removed.

It has now been removed.

Line 110. “molar ratio 7:3” on mass basis? Specify

It has now been modified accordingly.

Line 134. “91% ± 2.5, 97% ± 4 and 81%± 3.5 respectively” authors could organize their results in terms of encapsulation efficiency, mean size and zeta potential, using a table.

The parepared niosomes and disocomes are large vesicleswith a  µm scale not on a nano scale; therefore zeta potential is not applicable. Further size distribution graphs to the optimized formulations have been provided.

Figure 3 describes the shape of niosomes and gives an idea of their size and mean dimensions. However, could author indicate mean size and standard deviation for each sample produced?

The shapes were described and the particle size distributions for the optimized formulations have now been provided.

Line 245. “TNFa and NF-κB” a space more is present here.

Line 251. “TNFa, when comp” same problem

Line 253. “on TNFa and non-significant” same problem

Line 258. “on TNFa gene expression” same.

 Line spaces have been all adjusted

Materials and methods

Line 279. “recently published a new ciprofloxacin-Mannich derivative” why published? maybe “formulated”, “produced”?

We published the synthesis method and reported on the new CIP derivative

Line 310 “W denotes the amount of drug.” should be “the mass of drug”. It would be better to use concentration in this formula

It has now been modified accordingly

Line 327 “In vitro release”. In vitro should be in italique.

In vitro edited to italic formatting throughout the whole manuscript.

Line 335. this equation should become Nr. 4 and be put into the center.

Equation 4 should become equation 5 and so on.

 The equations numbering has been corrected accordingly.

Is it compulsory to indicate Section Nr. 5 on patents?

The statement  has now been modified accordingly

Reviewer 3 Report

  1. Line 48. Change: Being non-vascular Tissue receiving no blood  supply.

For: It is an avascular tissue, in which Macrophage mediated-tissue vascularization  

Diseases that can be associated with corneal neovascularization include inflammatory disorders, corneal graft rejection after transplantation, infectious keratitis, contact lens-related hypoxia, alkali burns, stromal ulceration, or limbal stem cell deficiency. 

Value include this reference:

Hadrian K, Willenborg S, Bock F, Cursiefen C, Eming SA, Hos D. Macrophage-Mediated Tissue Vascularization: Similarities and Differences Between Cornea and Skin. Front Immunol. 2021 Apr 7;12:667830. doi: 10.3389/fimmu.2021.667830. PMID: 33897716; PMCID: PMC8058454.

  1. Line 52. Add a paragraph like this, about the importance of corneal infections: over 1.5 million people worldwide will develop blindness from infectious corneal ulceration each year,3

,3Whitcher JP, Srinivasan M, Upadhyay MP. Corneal blindness: a global perspective. Bull World Health Organ. 2001;79(3):214–21. 

he relatively scarce attention given to infectious corneal ulceration does not reflect the impact of the condition on the most vulnerable, many of whom live in poverty.

  1. Line 65. Include something like thisInfectious keratitis is preventable and treatable: . The main etiologies of infectious corneal ulcers, including bacteria, fungi and parasites, are often clinically indistinguishable . One solution to the problem of infectious corneal ulceration may lie in the delivery of a simple, safe and effective community-based strategy, Early administration of topical drugs with a low level of bacterial resistance.

Ung L, Acharya NR, Agarwal T, Alfonso EC, Bagga B, Bispo PJ, Burton MJ, Dart JK, Doan T, Fleiszig SM, Garg P, Gilmore MS, Gritz DC, Hazlett LD, Iovieno A, Jhanji V, Kempen JH, Lee CS, Lietman TM, Margolis TP, McLeod SD, Mehta JS, Miller D, Pearlman E, Prajna L, Prajna NV, Seitzman GD, Shanbhag SS, Sharma N, Sharma S, Srinivasan M, Stapleton F, Tan DT, Tandon R, Taylor HR, Tu EY, Tuli SS, Vajpayee RB, Van Gelder RN, Watson SL, Zegans ME, Chodosh J. Infectious corneal ulceration: a proposal for neglected tropical disease status. Bull World Health Organ. 2019 Dec 1;97(12):854-856. doi: 10.2471/BLT.19.232660. Epub 2019 Nov 1. PMID: 31819296; PMCID: PMC6883276.

  1. Lines from 78 to 80: Very long paragraph, divide into two or three. Some sentences in the entire article are too long
  2. Line 105: correct : chemcial structure
  3. Line 184. Provide bibliographic or commercial reference: of ImageJ software.

Line 245. Correct this sentence: revealed that  all treated groups demonstrated statistically significant (P > 0.05) lower 

Author Response

Reviewer 3

  1. Line 48. Change: Being non-vascular Tissue receiving no blood  supply.

For: It is an avascular tissue, in which Macrophage mediated-tissue vascularization  

Diseases that can be associated with corneal neovascularization include inflammatory disorders, corneal graft rejection after transplantation, infectious keratitis, contact lens-related hypoxia, alkali burns, stromal ulceration, or limbal stem cell deficiency. 

The sentence has now been modified

Value include this reference:

Hadrian K, Willenborg S, Bock F, Cursiefen C, Eming SA, Hos D. Macrophage-Mediated Tissue Vascularization: Similarities and Differences Between Cornea and Skin. Front Immunol. 2021 Apr 7;12:667830. doi: 10.3389/fimmu.2021.667830. PMID: 33897716; PMCID: PMC8058454.

 The reference has now been added.

  1. Line 52. Add a paragraph like this, about the importance of corneal infections: over 1.5 million people worldwide will develop blindness from infectious corneal ulceration each year,3

,3Whitcher JP, Srinivasan M, Upadhyay MP. Corneal blindness: a global perspective. Bull World Health Organ. 2001;79(3):214–21. 

he relatively scarce attention given to infectious corneal ulceration does not reflect the impact of the condition on the most vulnerable, many of whom live in poverty.

  1. Line 65. Include something like thisInfectious keratitis is preventable and treatable: . The main etiologies of infectious corneal ulcers, including bacteria, fungi and parasites, are often clinically indistinguishable . One solution to the problem of infectious corneal ulceration may lie in the delivery of a simple, safe and effective community-based strategy, Early administration of topical drugs with a low level of bacterial resistance.

Ung L, Acharya NR, Agarwal T, Alfonso EC, Bagga B, Bispo PJ, Burton MJ, Dart JK, Doan T, Fleiszig SM, Garg P, Gilmore MS, Gritz DC, Hazlett LD, Iovieno A, Jhanji V, Kempen JH, Lee CS, Lietman TM, Margolis TP, McLeod SD, Mehta JS, Miller D, Pearlman E, Prajna L, Prajna NV, Seitzman GD, Shanbhag SS, Sharma N, Sharma S, Srinivasan M, Stapleton F, Tan DT, Tandon R, Taylor HR, Tu EY, Tuli SS, Vajpayee RB, Van Gelder RN, Watson SL, Zegans ME, Chodosh J. Infectious corneal ulceration: a proposal for neglected tropical disease status. Bull World Health Organ. 2019 Dec 1;97(12):854-856. doi: 10.2471/BLT.19.232660. Epub 2019 Nov 1. PMID: 31819296; PMCID: PMC6883276.

 It has now been added to the manuscript

  1. Lines from 78 to 80: Very long paragraph, divide into two or three. Some sentences in the entire article are too long
  2. Line 105: correct : chemcial structure

The word has now been corrected

  1. Line 184. Provide bibliographic or commercial reference: of ImageJ software.

Commercial reference has been provided

Line 245. Correct this sentence: revealed that  all treated groups demonstrated statistically significant (P > 0.05) lower 

The sentence has now been corrected.

Round 2

Reviewer 1 Report

The manuscript changes are in line with that recommanded. I suggest to accept the manuscript after thorough reading, as there are some mistakes in part of  the text. For instance : "Previous studies on ciprofloxacin indicated that encapsulation of ciprofloxacin in Previous studies on ciprofloxacin indicated that encapsulation of ciprofloxacin in Previous studies on ciprofloxacin indicated that encapsulation of ciprofloxacin in lipo somes has been more effective and reducing" The beginning of the sentence is repeated three times.

Author Response

The manuscript changes are in line with that recommanded. I suggest to accept the manuscript after thorough reading, as there are some mistakes in part of  the text. For instance : "Previous studies on ciprofloxacin indicated that encapsulation of ciprofloxacin in Previous studies on ciprofloxacin indicated that encapsulation of ciprofloxacin in Previous studies on ciprofloxacin indicated that encapsulation of ciprofloxacin in lipo somes has been more effective and reducing" The beginning of the sentence is repeated three times.

It has now been corrected with the clean version of the manuscript

Reviewer 2 Report

Authors provided a new version of their paper.

They addresses each points that I raised.

I only have a few minor issues.

Line 119. “in vivo” should be in italique. please revise in all the paper.

Line 157. “Previous studies on ciprofloxacin indicated that encapsulation of ciprofloxacin in”. this line is repeated in Lines 158 and 159.

Reference 40 is not reported. check it

Author Response

Authors provided a new version of their paper.

They addresses each points that I raised.

I only have a few minor issues.

Line 119. “in vivo” should be in italique. please revise in all the paper.

The word has been modified accordingly

Line 157. “Previous studies on ciprofloxacin indicated that encapsulation of ciprofloxacin in”. this line is repeated in Lines 158 and 159.

It has now been corrected in the clean version

Reference 40 is not reported. check

We have only 35 references